# Protein Substitutes in PKU; Their Historical Evolution

**DOI:** 10.3390/nu13020484

**Published:** 2021-02-02

**Authors:** Anne Daly, Sharon Evans, Alex Pinto, Catherine Ashmore, Anita MacDonald

**Affiliations:** Birmingham Women’s and Children’s Hospital, NHS Foundation Trust, Birmingham B4 6NH, UK; evanss21@me.com (S.E.); alex.pinto@nhs.net (A.P.); catherine.ashmore@nhs.net (C.A.); anita.macdonald@nhs.net (A.M.)

**Keywords:** phenylketonuria, protein substitute, amino acid, glycomacropeptide

## Abstract

Protein substitutes developed for phenylketonuria (PKU) are a synthetic source of protein commonly based on L-amino acids. They are essential in the treatment of phenylketonuria (PKU) and other amino acid disorders, allowing the antagonistic amino acid to be removed but with the safe provision of all other amino acids necessary for maintaining normal physiological function. They were first formulated by a chemist and used experimentally on a 2-year-old girl with PKU and their nutritional formulations and design have improved over time. Since 2008, a bioactive macropeptide has been used as a base for protein substitutes in PKU, with potential benefits of improved bone and gut health, nitrogen retention, and blood phenylalanine control. In 2018, animal studies showed that physiomimic technology coating the amino acids with a polymer allows a slow release of amino acids with an improved physiological profile. History has shown that in PKU, the protein substitute’s efficacy is determined by its nutritional profile, amino acid composition, dose, timing, distribution, and an adequate energy intake. Protein substitutes are often given little importance, yet their pharmacological actions and clinical benefit are pivotal when managing PKU.

## 1. Introduction

Amino acids are unique substrates providing nitrogen, hydrocarbon skeletons and sulphur [1]. They are essential precursors for the synthesis of proteins, peptides, and low molecular weight substances such as glutathione, dopamine, nitric oxide, and serotonin [1]. In phenylketonuria (PKU), dietary treatment was made feasible with the introduction of low/free phenylalanine synthetic proteins (protein substitutes), that have gradually advanced with time. In the 1950s, these were originally derived from protein hydrolysates, but in the 1970s, phenylalanine-free amino acids were introduced. Protein substitutes provide the building blocks of tissue proteins and their amino acids are essential for the synthesis of hormones, enzymes, and other cellular processes. Therefore, their composition and nutritional profile is fundamental, helping prevent neurological devastation, allowing normal growth and biosynthetic functions. The original technology for making protein substitutes was crude and limited but now precision manufacturing has improved their quality.

Although Følling [2] first identified phenylpyruvic acid in the urine of untreated children with PKU, it was Penrose who recognised that it was a genetic recessive disorder and named it phenylketonuria (PKU) [3]. He was also the first to try a dietary treatment based on fruit, sugar, olive oil, and vitamins, but this protein-free diet lacked essential phenylalanine and all other amino acids, resulting in malnutrition and so the treatment was abandoned [4]. Twenty years later, protein substitutes were introduced, but their central role in the management of PKU remains undervalued.

## 2. Early Studies

Følling and Penrose [5] both demonstrated that giving phenylalanine to a PKU subject increased the excretion of phenylpyruvic acid. The type of phenylalanine ingested as D or L isomers had different effects on phenylpyruvic excretion, with L phenylalanine leading to a greater production of phenylpyruvic acid. Similarly, in non-PKU subjects, L phenylalanine was the preferred metabolised substrate, with D and DL isomers leading to small amounts of phenylpyruvic acid but an absence when the L form was given due to its complete metabolism. From these studies, they concluded that phenylpyruvic acid excreted in PKU patients was due to an incomplete breakdown of phenylalanine. Tyrosine when administered had no effect on urine phenylpyruvic acid excretion, concluding this was metabolised normally. It was not until 1944 that Bernheim [6] demonstrated that the main metabolic pathway for phenylalanine was by parahydroxylation of phenylalanine to tyrosine. In 1953, Jarvis showed that it was the inability to perform this hydroxylation that resulted in phenylketonuria [7].

Penrose and Quastel [5] conducted a series of feeding studies where they found that by lowering the natural protein intake by >50% resulted in an immediate reduction in urinary phenylpyruvic acid in a patient with PKU. However, after the second day of treatment, urine phenylpyruvic acid re-appeared and increased over subsequent days. The authors noted a weight loss over the same time and hypothesised that catabolism led to the production of phenylpyruvic acid.

In 1951, a positive ferric chloride screening test in a symptomatic 2-year-old girl from Birmingham, UK, preceded the first successful dietary treatment in PKU. She was only the third child to be tested with the ferric chloride test at Birmingham Children’s Hospital [8]. In PKU, phenylpyruvic acid present in urine causes the characteristic greenish-blue colour reaction when a few drops of ferric chloride are added [9]. On presentation, she was unable to talk, walk, or engage with her surroundings; her mother waited for the doctor every morning outside the hospital laboratory as she refused to accept that there was no treatment for her daughters’ condition. Louis Woolf designed the first successful protein substitute formulation used in PKU. He was a chemist with a commercial background and had used hydrolysed casein to produce amino acids as a treatment for malnutrition after the Second World War. Cost was a priority in the post war years and protein hydrolysates were readily available and cheaper than pure amino acids. In 1949, he suggested that supplementation of carbon treated casein hydrolysate with appropriate amounts of missing amino acids (including a source of phenylalanine to prevent deficiency) could treat PKU. He was unable to convince his medical colleagues at Great Ormond Street (GOS) Children’s Hospital, London to try his proposed treatment. He recalls: “*At GOS, the suggestion floated like a lead balloon, I was told not unkindly that I should be devising new diagnostic tests, not dreaming up crazy treatments for conditions that everybody knew were untreatable*” [10]. In collaboration with Drs Bickel, Hickman, and Gerrard from Birmingham, a modification of Louis Woolf’s protein substitute was given to the 2-year-old child with PKU [11].

## 3. The First Protein Substitute

Casein was hydrolysed by boiling it for several hours with concentrated hydrochloric or sulphuric acid to produce a thick black amino acid liquid. This solution was neutralised with sodium hydroxide and then purified by the addition of activated carbon and finally filtered to produce a clear solution of amino acids. This solution contained phenylalanine, which was removed by a second filtration method using activated charcoal. This removed the aromatic amino acids: phenylalanine, tryptophan, and tyrosine (although a small residual amount of phenylalanine was detectable). To nutritionally improve the protein substitute carbohydrate, fat, vitamins and minerals were added, together with tryptophan and tyrosine [11]. This unpalatable solution was then mixed with sugar, wheat starch, double cream and water and given as a formula to infants. In older children, it was either flavoured with tomatoes and given as a soup [12], made into a blancmange with sugar, margarine and wheat starch [13] or mixed with vegetable oil, and sugar and flavoured with artificial flavourings [14].

The production of the original formula was difficult and time consuming and had to be done in a cold room or it would deteriorate. The black charcoal covered everything and as the first formula was prepared, the sight of Dr. Bickel wrapped in layers of jumpers topped by a charcoal smudged lab coat became a common sight [8]. Woolf identified that a small amount of phenylalanine should be added to the formula as it was an essential amino acid [15]. He stressed the need for careful monitoring, and he was also the first to propose treatment for life in PKU [12,16,17].

Phenylalanine-free amino acids as a protein substitute for PKU were first tried in the USA in the 1950s [18], but had to be abandoned most likely due to the pure amino acid mixtures causing vomiting.

## 4. Amino Acid Requirements

In the early stages of making the protein substitute, the exact amino acid composition of the casein hydrolysate was unknown. Casein was low in sulphur containing amino acids and cysteine was also partly removed by hydrolysis and charcoal filtration. Bickel [19] suggested adding L cystine to the hydrolysate and Woolf proposed the addition of DL methionine [20].

The amounts of tryptophan and tyrosine added to the first protein substitutes were determined by amino acid requirements established in the early 1950s; this knowledge was pioneered by Rose [21,22,23,24,25], Holt and Snyderman [26,27]. It was estimated that an adult man required 1000 mg/day of L phenylalanine to maintain nitrogen equilibrium or 300 mg/day if tyrosine was provided [24]. Synderman [28] suggested that 90 mg per kg/day of L phenylalanine was needed by an infant, but this was reduced to 25 mg/kg per day if sufficient tyrosine was supplied. Other essential amino acid requirements were estimated from the work of Rose, Leverton [23,29] and Swendseid [30]. In Woolf’s [12] original formula, 25 mg/kg/day of L tryptophan and 25 mg/kg/day of L methionine were given in addition to 50 mg/kg/day of L tyrosine, a surrogate essential amino acid in PKU (Table 1).

There were challenges when administering the artificial diet in PKU [11]. The first child to start treatment was admitted to hospital for 6 weeks. The musty smell associated with PKU disappeared, plasma and urinary phenylalanine concentrations returned to normal, and there was a negative ferric chloride test when the diet was commenced. However, the child lost weight and within 5 days of treatment, plasma tyrosine concentrations were un-recordable (with a change in hair pigmentation), and plasma phenylalanine was raised. Tyrosine (1.5 g/daily) was added, correcting the low plasma tyrosine concentrations and temporarily arrested weight loss. However, after a further 3 weeks, aminoaciduria was noted, and in the fifth week, blood phenylalanine increased and phenylpyruvic acid reappeared in the urine; this was associated with weight loss, vomiting, and the child was described as unwell. These observations were important, highlighting that tyrosine became an essential amino acid in PKU as a consequence of the biochemical block in converting phenylalanine to tyrosine. Aminoaciduria, a result of weight loss and catabolism due to phenylalanine deficiency, led to an increase in catabolism and a subsequent increase in blood phenylalanine. Adding a measured amount of phenylalanine back into the diet (typically 250–500 mg or equivalent to around 5–10 g protein/day) increased plasma and urine concentrations, but to levels significantly below pre-treatment concentrations. Laboratory analysis was laborious, each blood test was analysed in duplicate and the production of a chromatogram took 3 days of intensive labour [8].

After 6 months of treatment, this child made remarkable progress; followed by a cascade of successful case studies. Woolf [17] reported 3 cases, with a further publication of 10 cases in 1958 [12], in which children were treated from the age of three weeks to 5 years of age. Armstrong and Tyler [31] reported the treatment of five children, and Armstrong and Binkley in 1956 [32] followed the progress of an infant starting treatment at 40 days of age. All reported that a low phenylalanine diet, supplemented with a low phenylalanine protein hydrolysate corrected the major biochemical abnormalities.

It was also established that sufficient carbohydrate and fat (including a source of linoleic acid) was necessary to prevent protein catabolism [20,33,34,35]. Woolf reported that the daily intake of hydrolysate should be high correcting for the inefficient utilisation of the amino acids [12].

## 5. Commercial Protein Hydrolysate Preparations

Production of the hydrolysate moved from hospital laboratories to commercial production in late 1953/early 1954. In Europe, Cymogran 1954/5 (Allen and Hanbury, London, UK), XP Albumaid (1960) (Scientific Hospital Supplies, Liverpool, UK) and Minafen (designed for infants in 1955), (Trufood Ltd., Guildford, UK) were developed and the US produced Lofenalac (Mead Johnson, Chicago, IL, USA) in 1958. In the spirit of commercial interest, Trufood and Allen and Hanbury agreed to share production with one company making an infant substitute Minafen (Trufood)and the other (Allen and Hanbury) a preparation for older children Cymogran. Limited practical instructions were provided on how to reconstitute these formulas and families had to weigh the prescribed powder. The main difference between hospital and factory production was the use of ion exchange resins to separate phenylalanine, dispensing with the sodium hydroxide and carbon filtration. These synthetic filters consisted of microbeads from resin or polymers, allowing the separation and purification of the hydrolysed casein. These products were supplemented with variable amounts of vitamins, minerals, carbohydrate and fat.

## 6. The First UK PKU Guidelines

In 1960, the UK Ministry of Health [9] provided guidelines on screening and early detection of PKU, together with recommendations on optimal blood phenylalanine concentrations and provision of protein substitutes. They proposed screening by the ferric chloride test at 4–6 weeks of age (which was later replaced by the Guthrie method in 1969 [36]). To prevent phenylalanine deficiency, a target blood phenylalanine concentration slightly above normal was recommended (90–120 mmol/L), with blood phenylalanine monitoring done twice weekly until stability was achieved, and then weekly or monthly monitoring was required.

In infancy, a protein substitute, formulated and reconstituted similar to regular milk-based infant formula was recommended. A second protein substitute with a lower energy content was advocated for older children.

## 7. Nutritional Deficiencies with Early Protein Substitutes

In the early history of treating PKU by diet, there were concerns about ‘over- treating’ patients and maintaining very low phenylalanine blood concentrations. Nutritional deficiencies, malnutrition, and even death were linked to dietary treatment [37]. In the 1960s, severe skin rashes in babies on Minafen (Allen and Hanbury Ltd., London, UK) were reported [38,39,40]. Woolf [12] described a child with faltering growth and hair loss when acetyl DL tryptophan was accidentally given instead of DL tryptophan; stopping the acetyl derivative immediately reversed the symptoms. Studies in animals fed synthetic low phenylalanine diets [41] led to the addition of choline, riboflavin, folic acid, and vitamin E to the hydrolysate preparations. Two reports of folic acid deficiency were described [42,43], one child had megaloblastic anaemia due to folic acid deficiency exacerbated by vomiting and poor feeding and subsequently died. Hypoglycaemia was also reported in two cases [44].

## 8. Amino Acid Preparations

In the late 1960s, commercial amino acid formulas were made from pure crystalline amino acids by fermentation of bacteria. They were manufactured by the Japanese at an affordable cost [45]. The first product for PKU was Aminogram Food Supplement [46,47], which had several advantages, compared to hydrolysed formulas including improved taste and a lower daily volume, with an amino acid composition that could be easily adapted for the treatment of other aminoacidopathies such as maple syrup urine disease, homocystinuria, and tyrosinaemia type 1.

Manz [48] reported anorexia and vomiting in some infants given amino acid preparations. Metabolic acidosis was observed when the preparations contained amino acids in the form of hydrochloride salts or when the ratio of sulphur containing amino acids was too high, leading to higher urinary pH and increased renal net acid excretion. Modifications in the amino acid preparations normalised the renal net acid excretion and acidosis.

The early commercial preparations of L amino acid substitutes required separate supplementation with vitamins and minerals and careful monitoring of nutritional status was essential. These vitamin and mineral supplements were commonly deficient in molybdenum, chromium, selenium, and pantothenic acid [49,50].

**L-amino acid substitutes nutritionally complete:** In 1980, the first UK amino acid preparation supplemented with carbohydrate, vitamins, mineral, and trace elements and designed for children over 1 year of age with PKU was manufactured. It was flavoured for improved taste and palatability. In 1988, a similar product (but also with added taurine and carnitine), but formulated for children over the age of 8 years and suitable for maternal PKU, was introduced [51]. From the 1990s, further advances were made in the nutritional formulations, taste, and presentation of protein substitutes (Table 2). Although selenium supplementation was added to protein substitutes from the late 1980s, many countries were wary about adding selenium to protein substitutes due to concerns about its toxicity which had been responsible for deaths in man and animals and was referred to as the ‘essential poison’ [52]. Consequently, this led to reports of many cases of biochemical selenium deficiency [53,54].

It was also established that the fat intake of children with PKU was low [55] and n-3 long chain polyunsaturated fatty acid status was sub-optimal [56]. This led to the addition of essential fatty acids to protein substitute powders designed for children [57]. Around the same time, long chain polyunsaturated fatty acids were added to infant protein substitutes in 2000; in a double blind randomised study, infants received either a formula with or without a supplemented fat blend of long chain polyunsaturated fatty acids (LC-PUFA). The results clearly showed the benefit of supplementation [58] and led to the addition of LC-PUFA to other products designed for older children.

Over the years, there has been much endeavor to ensure that protein substitutes meet changing nutritional trends and accommodate nutritional requirements according to life stage. In 2011, the first phenylalanine-free infant formula containing a specific mixture of prebiotic oligosaccharides was introduced [59]. This helped maintain levels of bifidobacteria and lower stool pH in infants with PKU. There is concern about increasing obesity rates in the PKU and non PKU population, so many recent protein substitutes introduced for children, teenage, and adults with PKU have a lower carbohydrate and energy composition [60]. Impact on lowering obesity rates has not yet been proven.

The nutritional adaptation of protein substitutes in order to gain clinical benefit is an area likely to grow in the future. Recently, specific nutrient combinations (containing uridine monophosphate, docosahexaenoic acid, eicosapentaenoic acid, choline, phospholipids, folic acid, vitamins B12, B6, C, and E, and selenium) have been studied in PKU mice to examine the impact on synaptic deficits in PKU [61]. The specific nutrients are precursors and cofactors for the synthesis of phospholipids thought to be beneficial in improving the neurotransmitter/synaptic changes in PKU. This combination of nutrients has been shown to have a benefit on synapse formation, morphology, and function in mouse models of Alzheimer’s disease so it may be an important nutritional adaptation of protein substitutes for older patients [62].

## 9. Choice of Protein Substitutes

The choice, composition, and presentation of protein substitutes have expanded at fast moving rates since the turn of the century. This time was associated with the evolution of pre-packaged and premeasured products, which has not only improved convenience but also accuracy, adherence, and ease of protein substitute prescription for clinicians.

**Spoonable low volume protein substitutes:** An innovative substitute designed for young children with PKU was produced in 2000, based on phenylalanine free amino acids and starch to which a small amount of water was added, forming a gel/paste that was similar in consistency to a weaning food [60]. This low volume, fat free, lower calorie, more concentrated amino acid substitute had the advantage of allowing transition onto a second stage product from the age of 6 months, in line with complementary feeding. It was presented in premeasured sachets (dispensing with the need for large tins of formula), was easy to prepare, with a good consistency and acceptable taste. Normal infant feeding behaviour, teething, and intercurrent infection can lead to its rejection in late infancy so perseverance and a consistent approach is needed by parents [63,64].

**Ready to drink liquid protein substitutes:** In 2005, ‘ready to drink’ flavoured phenylalanine free amino acid pouches were introduced. These small volume, lower energy protein substitutes were convenient and compact, allowing greater independence for children and teenagers. Patients were less self-conscious taking a liquid drink compared to a powdered preparation [65]. The pharmacological efficacy of these lower volume substitutes did not compromise nutritional biochemistry or phenylalanine concentrations, which remained the same or improved [60]. One potential problem was abdominal discomfort (constipation/diarrhoea), attributed to the hyperosmolar concentration of the lower volume protein substitutes [66,67], and like all concentrated amino acid products, they should be administered with additional water.

**Protein substitute tablets:** Amino acid tablets and modular systems were also introduced around 2000. Modular systems are when a combination of amino acid tablets, capsules, liquids, powder, or bars of amino acids are used to provide daily protein substitute requirements, allowing flexibility of choice. In a randomised crossover study, it was shown that subjects with PKU successfully took at least 40% of their protein substitute as tablets, with an improvement in adherence and significantly lower blood phenylalanine concentration [68]. The quantity of tablets to meet protein requirements was around 70/day, and they were not nutritionally complete, requiring extra supplementation with vitamins and minerals. They provide an alternative for older children and adults who struggle taking conventional protein substitute. Micro-tablets of amino acids have since been introduced.

**Caseinglycomacropeptide with amino acids (CGMP-AA)**: CGMP-AA was introduced in the UK in 2017, although first used in the USA as a protein substitute for PKU in 2008 [69]. CGMP is purified from whey by anion exchange chromatography, but the final product does contain residual amounts of aromatic amino acids including phenylalanine [70]. CGMP-AA is different from amino acid substitutes; approximately 40% of the product is composed of amino acids, with the rest as a bioactive peptide; based on a macropeptide, they are associated with improved taste and palatability [71].

**Slow-release protein substitute:** A prolonged release product was first developed in 2014 [72] but there was little supporting published data demonstrating its effectiveness. In 2018, a slow release preparation containing amino acids coated with ethyl cellulose and alginate was introduced. Based on physiomimic technology, the bitter taste and smell of amino acids was improved, and as this product is not mixed with fluid, it does not have an osmolality. Most importantly, the technology prolongs the release of amino acids into the systemic circulation. Animal and human kinetic studies demonstrate a reduced peak concentration of amino acids. This new technology suggests a physiological absorption of amino acids similar to natural protein [73,74]. In a short-term observational study using prolonged amino acids in subjects with PKU, it was well tolerated, with fewer gastrointestinal symptoms and no change in blood phenylalanine concentrations [75].

## 10. Pharmacological Importance of Protein Substitutes

The amount of protein equivalent (g/kg) from protein substitutes affects blood phenylalanine control [76,77,78,79]. As early as 1961, an observational study performed by O’Daly [80] showed protein substitutes significantly lowered blood phenylalanine concentrations. Further studies have shown that phenylalanine tolerance is increased when total protein intake from a protein substitute is increased [79,81].

Protein substitutes have an important function at the blood brain barrier. Large neutral amino acids including phenylalanine compete for LAT1, a large neutral amino acid transporter allowing entry of amino acids into the brain [82]. Phenylalanine has a particularly high affinity for LAT1 and protein substitutes are the only source of competitive large neutral amino acids necessary to prevent excess phenylalanine entering the brain. These pharmacological effects of ingesting an amino acid rich formula are frequently neglected and given little scientific credence, and yet they have a significant impact on phenylalanine metabolism and long-term physical and neurological outcome. The gut also controls the absorption of amino acids across the epithelial membrane. Phenylalanine is transported as a carrier mediated sodium dependent process which requires energy. Similar to the blood brain barrier, large neural amino acids are transported in the gut by LAT1, also known as SLC5A7, for which phenylalanine has a high affinity [83,84].

## 11. Protein Substitute Requirements

Human requirements for each amino acid are specific to age, metabolic demands (immune/neuromuscular), and growth rate (protein deposition) [76,85,86]. For protein synthesis to occur, all the amino acids should be available; absence of one leads to the cessation of synthesis [1]. Snyderman [87] reported that the complete withdrawal of phenylalanine from the diet in a normal infant led to the depression in several other amino acids, the most prominent being tyrosine, hence the importance of tyrosine supplementation. Woolf showed that the nitrogen content of the artificial substitute was not an exchange for natural protein; the hydrolysate contained less nitrogen, was rapidly absorbed from the gut with greater oxidation and urinary amino acid losses [88,89], therefore sufficient product was needed to meet nitrogen requirements.

A protein substitute intake that just meets minimum WHO requirements (WHO/FAO/UNU 2007) [90] may result in ‘latent’ catabolism, leading to body tissue breakdown, increasing phenylalanine concentrations. Protein utilisation is enhanced by a supply of carbohydrate and fat [91,92] further illustrated in a randomised controlled study in PKU subjects by MacDonald [93] and supported by Illsinger [94].

The European PKU Guidelines recommend that the total protein intake should supply 40% more than the FAO/WHO/UNU safe levels of protein intake [95]. However, this amount is arbitrary and unconfirmed by research [67]. A collaborative study [96] involving 63 European and Turkish IMD centres concluded that the amount of total protein prescribed by different European countries was not uniform. All centres gave higher protein equivalents than the recommended 2007 WHO/FAO/UNU [90] safe levels of protein intake with Western European centres prescribing less total protein then other European regions.

To maximise the utilisation of amino acids and minimise the variation in phenylalanine concentrations, protein substitutes should be taken frequently, a minimum of three times a day. MacDonald [97] demonstrated that the greater the amount of protein substitute consumed between waking and 4 p.m., the greater the decrease in phenylalanine concentrations. Likewise, when protein substitute was given 4 hourly for 24 h, there was a marked stablisation in phenylalanine concentrations, reducing phenylalanine variability [98].

Tyrosine, a precursor of catecholamine neurotransmitters (dopamine, norepinephrine, and epinephrine), thyroxine, and melanin, is an essential amino acid in PKU due to the limited or absent hydroxylation of phenylalanine. It is hydrophobic and the absolute quantities added to protein substitutes are not defined. Indicator amino acid oxidation studies [99] suggest tyrosine should provide 19 mg/kg/day, although current protein substitutes provide approximately 5 times above current recommendations. The importance of tyrosine was recognised by the Report of the Medical Research Council working party on PKU [100], which recommended that protein substitutes should be nutritionally complete and contain 100–120 mg/kg/day of tyrosine.

## 12. Protein Substitute Administration

In the early history, the practicalities of administering an acid based hydrolysed unflavoured product were particularly challenging. Bentovim [46] described the struggles families faced trying to persuade children to take the acid tasting formula: the large daily volume that needed to be consumed, regular vomiting, refusal to eat permitted food due to negative associations with the substitute, the bad smell, and lack of palatability. Furthermore, children experienced isolation and psychological difficulties particularly in the school years. This was one of the factors leading to diet cessation as early as 6 years [101,102,103]. An extract from the *Cork Examiner* describes the struggles faced by one family adapting to the news that their two children had been diagnosed with PKU and the dietary changes and challenges made to improve their neurological outcome [104].

Despite the advances in technology, almost all protein substitutes have a strong taste and odour and are associated with poor palatability and breath odour. They are a burden to patients as they must be consumed a minimum of three times daily and spread evenly throughout the day. Ford [66] reported 293 of 631 participants with PKU (39% of adults, 11% of children) either did not take protein substitute or took less than their prescribed amounts.

Verbatim extract from study: *Our greatest struggle is getting our son taking his protein substitute. He refuses to take it and it can take up to 45 min for him to finish one with a lot of upset*.

Evans [105], in a case control study in PKU children of weaning age, highlighted the stress, anxiety, and struggles associated with protein substitute administration. Maternal anxiety regarding child rejection of protein substitute increased with time peaking at 12–24 months. Similarly, in 2016, MacDonald [106] reported in 114 children with PKU, dietary management was associated with a considerable time and financial burden for caregivers, with much time spent supervising protein substitute intake.

## 13. Conclusions

In PKU, the early pioneers understood the physiological importance of protein substitutes. They stressed the need for a balanced amino acid profile, for even administration throughout the day, together with an adequate energy intake and dietary treatment for life. Although these principals remain unchanged 70 years later, each decade has witnessed improvements in the delivery and nutritional composition of protein substitutes, which remain of fundamental importance in the treatment of PKU. Further changes are needed, to deliver improved taste and odour-free products, with the properties of natural protein delivering a stable chemical environment associated with optimal physiological function and patient tolerance.

An extract from the *Irish Cork Examiner* describes the struggles of a family diagnosed with PKU in 1959, the son aged 4 and the daughter 2½ years old. This extract describes the determination, sacrifice, hardship, and success against the better judgement of expert advice.

25 October 1962. Phenylketonuria: A story of heartbreak and hope.


*“Treatment might help your daughter” he said “but for your son detection has come too late.” He would deteriorate so much that at a later stage institutional care was inevitable. No one had attempted treatment on a child over 2 years. But the specialist was willing to give my little girl a trial. I pleaded for both of them not knowing the terrible struggle this entailed.*



*The boy was difficult, backward and had no speech while his sister could neither walk or talk and was unable to sit up alone and was extremely difficult to manage. Those first months of 1959 were a nightmare from which there was no awakening. The introduction of the unpalatable diet and the cessation of stews, broths and chocolate sundaes brought tears and tantrums. How I dreaded the ice cream vendors that first summer and the laughing lolly licking youngsters who stood on our corner. The synthetic protein (Minafen) was unpleasant to take, but I have found it can be disguised reasonably well in savouries and cookies.*



*Each child is allowed approximately 270 mg of phenylalanine per day according to body weight. If the child were to have 1oz of porridge oats this would cover 241.5 mg of their daily allowance, whereas 1oz of tomatoes would only represent 5 mg, so planning meals for them was at one time a highly complicated business. Now I possess a simplified chart of foods with a low phenylalanine content and by drawing on almost every cook book in print for ideas I have complied my own Cook Book for Phenylketonurias. Special gluten free flour must be used for bread making and Kosher margarine replaces butter, bread making with wheat starch was a different matter “Neolite or just plain leather “was my husband’s query at my first attempt. Meals for ourselves present a real problem. It is so difficult to take a hearty T bone steak or peach meringue while two pair of eyes watch with longing. Meals out are impossible as is home entertainment, but it has been so worthwhile, the little girl unable to talk or walk takes some chasing and her speech is coming slowly. Responsiveness and alertness have taken place in slow but sure degrees. My son now 7 has made remarkable progress benefiting from a normal education.’*



*Although detection of PKU and treatment soon after birth is essential for complete recovery we have proved beyond all doubt that much can still be done.*


## Figures and Tables

**Table 1 nutrients-13-00484-t001:** The original composition of the first protein substitute designed by Louis Woolf (1958).

Product	Daily Intake
Casein hydrolysate	**24 g**
DL-tryptophan	**1 g**
L-tyrosine	**2 g**
DL-methionine	**1 g**
Sucrose	**90 g**
Cows milk	**0–200 mL**
Double cream	**85 mL**
Calcium hydrogen phosphate	**0.71 g**
Potassium chloride	**0.65 g**
Sodium chloride	**0.016 g**
Magnesium sulphate	**0.165 g**
Sodium citrate	**0.177 g**
Potassium iodide	**0.00013 g**
Citric acid	**0.08 g**
Water	**850 mL**
**Vitamins and minerals**	
Choline chloride	**100 mg**
Inositol	**216 mg**
Vitamin B12	**4 μg**
Aneurine hydrochloride (vitamin B1)	**0.5 mg**
Riboflavin	**0.5 mg**
Pyridoxine	**0.33 mg**
Nicotinamide	**3.33 mg**
Ascorbic acid	**40 mg**
α-Tocopherol	**0.33 mg**
Acetomenaphthone (vitamin K)	**0.5 mg**
Biotin	**0.17 mg**
Folic acid	**0.35 mg**
Vitamin A	**3000 iu**
Vitamin D	**500 iu**
Zinc sulphate	**0.0014 g**
Ferrous sulphate	**0.15 g**
Manganous sulphate	**0.0008 g**
Cupric sulphate	**0.003 g**

**Table 2 nutrients-13-00484-t002:** Introduction of protein substitutes.

**Laboratory produced preparation**	
**1952**	**Hydrolysed casein, powdered preparation, nutritionally incomplete, low phenylalanine** Addition of tyrosine, tryptophan and methionine, carbohydrate, fat, vitamins and minerals
**Commercial produced preparations**	
**1954**	**Hydrolysed casein, powdered preparation, nutritionally incomplete, low phenylalanine** Trufood (infant product), Allen and Handbury/Mead Johnson (older children)
**1960**	**L amino acid, powdered preparation, nutritionally incomplete phenylalanine free** Powell and Scholfield powdered preparation taken as a drink
**1980**	**L amino acid, powdered preparation with added carbohydrate and fat nutritionally complete, phenylalanine free** Powdered flavoured preparation taken as a drink
**1988**	**L amino acid, powdered preparation with added carbohydrate and fat nutritionally complete, phenylalanine free** Powdered preparation taken as a drink for >8 years and adults
**1988**	**L amino acid, powdered preparation, with added carbohydrate and fat, nutritionally complete designed for infants, phenylalanine free** Powdered preparation used for infants 0–12 months
**2001**	**L amino acid, powdered preparation with added carbohydrate and essential fatty acids, nutritionally complete, phenylalanine free** Powdered flavoured preparation taken as a drink
**2001**	**L amino acid powdered preparation, spoonable paste added carbohydrate, fat free, nutritionally complete, phenylalanine free** Powdered flavoured preparation taken as a spoonable paste
**2002**	**L amino acid tablets, nutritionally incomplete, phenylalanine free** Amino acid tablets
**2003**	**L amino acid powdered preparation, low carbohydrate, no fat, nutritionally complete, phenylalanine free** Powdered flavoured preparation taken as low volume drink or spoonable paste for children >8 years
**2006**	**L amino acid ready to drink preparation, low carbohydrate no fat, nutritionally complete, phenylalanine free** Ready to drink flavoured liquid
**2008**	**Casein macropeptide with L amino acids, essential fatty acids, nutritionally complete, low phenylalanine** Powdered preparation made into a low volume drink
**2008**	**Casein macropeptide with L amino acid nutritionally complete with essential fatty acids, low phenylalanine.** Powdered preparation make into a low volume drink
**2018**	**Slow release L amino acid preparation, carbohydrate and fat free, nutritionally complete, phenylalanine free** Micro tablets, made from L amino acids coated with ethyl cellulose and alginate which slowly release the L amino acids

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
