# Peer review of "Protein Substitutes in PKU; Their Historical Evolution"

_nutrients, 2021, doi:10.3390/nu13020484_

Round 1

Reviewer 1 Report

I very much enjoyed reading this interesting and well researched history of the development of PKU substitutes.

I believe it will be valuable to practitioners who work in, but also out of the area of PKU. The details are fascinating.

I appreciate that not all details can be included in such a review, although alluded to throughout the text, I believe that some additional emphasis and clarification could be included in the following areas, particularly for those who do not work in PKU:

Explain what the purpose of the ferric chloride test more explicitly when first mentioned.

The composition and administration of substitutes has evolved as we have understood more about the PKU and outcomes, as well as research into the efficacy of the substitute itself as a key to control. Thus it is seen as both a ‘medicine’ with a therapeutic effects as well as nutritional providing essential protein and nutrients. As more is learned about the requirements and outcomes in PKU and along with the recommendation for diet for life, this has influenced the development of more novel products to better meet the needs all life stages. For example rising rates of obesity in both the general population as well as for the PKU population is more of a concern in modern times compared to the malnutrition of early days.

In addition, as the development of low specialised food products has evolved broadening the diet of patients, this has heavily influenced the composition of supplements to ensure energy balance is maintained and to address subsequent changes in appetite.

We have also viewed substitutes with more of a developmental focus such as those used in weaning.

A little more detail on the basis of dosing recommendations which are important for protein utilisation, phenylalanine control and to meet nutritional requirements. This requires them to be routinely assessed depending on age, growth, control and compliance.

The evolution of pre-packaged and premeasured products has increased convenience but also accuracy, compliance and ease of prescription for clinicians

Author Response

Dear Editor 

Please find the revised manuscript and letter to the Editor as requested with the responses to the reviewers.

I have downloaded two documents 

  1. Letter of response
  2. Corrected manuscript

Thank you 

Anne Daly

Reviewer 2 Report

Dear Anne, Sharon, Alex, Catherine and Anita, 

I read your review "protein substitutes in PKU; their historical evolution" with much interest and pleasure. I have no major (or minor) comments or addition to your historical overview of PKU protein substitutes.

Author Response

Dear Editor

Please find the revised manuscript and the point by point letter to the reviewers

Thank you Anne Daly

Hopefully both have been downloaded below

  1. Letter
  2. Manuscript
